# Circular RNAs and Untranslated Regions in Acute Myeloid Leukemia

**DOI:** 10.3390/ijms24043215

**Published:** 2023-02-06

**Authors:** Shih-Chia Yeh, Faith J. F. Cheong, Yvonne Tay

**Affiliations:** 1Cancer Science Institute of Singapore, National University of Singapore, Singapore 117599, Singapore; 2NUS Centre for Cancer Research, Yong Loo Lin School of Medicine, National University of Singapore, Singapore 117599, Singapore; 3Department of Biochemistry, Yong Loo Lin School of Medicine, National University of Singapore, Singapore 117596, Singapore

**Keywords:** circular RNA, 3′untranslated region, alternative polyadenylation, 3′UTR splicing, single nucleotide polymorphism

## Abstract

Before the advent of next-generation sequencing, research on acute myeloid leukemia (AML) mostly centered on protein-coding genes. In recent years, breakthroughs in RNA sequencing technologies and whole transcriptome analysis have led to the discovery that approximately 97.5% of the human genome is transcribed into non-coding RNAs (ncRNAs). This paradigm shift has led to an explosion of research interest in different classes of non-coding RNAs, such as circular RNAs (circRNAs) as well as non-coding untranslated regions (UTRs) of protein-coding messenger RNAs. The critical roles of circRNAs and UTRs in AML pathogenesis have become increasingly apparent. In this review, we discuss the cellular mechanisms of circRNAs and summarize recent studies that reveal their biological roles in AML. Furthermore, we also review the contribution of 3′UTRs to disease progression. Finally, we discuss the potential of circRNAs and 3′UTRs as new biomarkers for disease stratification and/or the prediction of treatment response and targets for the development of RNA-directed therapeutic applications.

## 1. Introduction

Acute myeloid leukemia (AML) is characterized by the clonal expansion of immature blast cells (>10%) in the bone marrow and peripheral blood [1], which results in ineffective erythropoiesis, myeloid differentiation blocking and bone marrow failure [2]. Approximately 60–70% of AML adult patients achieve first complete remission (CR) with the advancement of chemotherapy, hematopoietic stem cell transplantation, molecular targeted therapy and immunotherapy [3,4]. Standard induction therapy is known as the 7+3 regimen, which consists of taking cytarabine for 7 days, along with anthracycline on each of the first 3 days. The CR rate of this first-line treatment is 60–80% in younger patients (<65 years old) and 40–60% in patients over 65 years old [5,6,7]. However, nearly 50% of patients cannot achieve CR, and these patients rely on alternative treatment strategies such as FLT3 inhibitors which target ITD and/or TKD mutations in the FLT3 receptor [8]. The initial classification of AML in 1985 was divided into subtypes M0 through M7 based on the morphological appearance of myeloid blasts [9]. In the revised 4th edition of the WHO classification published in 2017, AML is classified into six categories: AML with recurrent genetic abnormalities, AML with myelodysplasia-related changes (MRC), therapy-related myeloid neoplasms (t-MN), AML not otherwise specified (NOS), myeloid sarcoma and myeloid proliferation related to Down syndrome (DS) [10]. The new categorization is based on advanced high-throughput sequencing and gene expression data on hematologic malignancies [11]. High-throughput sequencing has led to the discovery of multiple non-coding regions, which could represent novel biomarkers and/or therapeutic vulnerabilities that warrant further investigation.

At present, protein-coding genes annotated in the Human Genome Project account for less than 2.5% of the human genome [12]. The remaining transcripts in the genome are considered non-coding RNAs (ncRNAs), comprising small (<200 nucleotides) ncRNA species such as microRNAs (miRNAs) and PIWI-interacting RNAs (piRNAs), long (>200 nucleotides) ncRNAs (lncRNAs) and circular RNAs (circRNAs) [13].

CircRNAs are single-stranded RNAs with closed-looped secondary structures produced from precursor mRNA (pre-mRNA) back-splicing. They were originally classified as ncRNAs since they are devoid of 5’ caps and 3’ end poly(A) tails necessary for cap-dependent translation [13,14]. However, RNA sequencing coupled with polysome profiling has revealed the translational potential of several circRNAs via different cap-independent mechanisms, including IRES (internal ribosome entry site), MIRES (m^6^A internal ribosome entry site) and rolling translation [15]. 

Many ncRNAs are involved in the pathogenesis of various diseases; for instance, miRNAs, which are short ncRNAs approximately 20-24 nucleotides in length [16]. They post-transcriptionally regulate target transcripts by binding to response elements and regulating the transcript stability or translational efficiency of messenger RNAs (mRNAs) [17]. Additionally, miRNAs possess additional functions independent of their post-transcriptional regulation of transcript expression. For example, they can function as transcriptional regulators: the 5' seed sequences of several miRNAs, such as miR-H3, mir-138, mir-92a and mir181d, have been shown to bind to DNA promoters [18]. MiRNAs have been shown to modulate AML disease development, e.g., cellular proliferation, survival and differentiation [19]. PiRNAs are single-stranded RNAs about 26-31 nucleotides in length that interact with PIWI proteins to form the piRNA silencing complex (piRISC) in most animals. Although studies have characterized the role that piRNAs play in solid tumors [20], little is known about their potential function in blood cancers. 

LncRNAs can be mainly classified into three groups based on their genomic location: antisense lncRNAs [21], enhancer lncRNAs [22] and intergenic lncRNAs [23]. Natural antisense lncRNAs are transcribed from the opposite strand of genes. They can mediate sense gene expression and affect the expression of other genes through *cis*- and *trans*-acting regulation, respectively [24]. Enhancer RNAs are another group of lncRNAs transcribed from enhancer regions and are essential for enhancer activities [22]. In addition, they play an important role in transcriptional regulation through their interactions with transcription factors, cofactors and RNA-binding proteins (RBPs) [25]. Intergenic lncRNAs are defined as autonomously transcribed lncRNAs that do not overlap with annotated coding genes. The functions of intergenic lncRNAs include the remodeling of chromatin and genome architecture, RNA stabilization and transcription regulation [26]. Although many of these ncRNAs play vital roles in the disease, we will not discuss them as they have either been recently reviewed in the literature [27,28] or are not well studied in AML. In this review, we will thus focus more on circRNAs. 

Other than distinct species of ncRNAs, mRNAs comprise central protein-coding regions flanked by non-coding 5’ and 3’ untranslated regions (UTRs) that control key facets of mRNA post-transcriptional processing and have been shown to play important roles in AML pathogenesis. Moreover, most mammalian genes generate mRNAs with alternative 3’UTRs via processes such as alternative polyadenylation (APA) [29] and 3’UTR splicing [30]. The occurrence of 3'UTR APA takes place due to the existence of multiple polyadenylation signals (PASs) within the same transcript and results in differential sequence information at the 3’ end. It is regulated by various *cis*-regulatory elements such as the PAS, which generally has a consensus sequence of AAUAAA or other similar sequence variants. As the PAS recruits cleavage and polyadenylation specificity factors (CPSFs), which are *trans*-regulators, it initiates the 3’UTR cleavage and polyadenylation process. As 3’UTRs are frequently associated with RBPs, lncRNAs and miRNAs that act as post-transcriptional regulators, APA-mediated changes in 3’UTR lengths can result in differential mRNA stability, translational efficiency and protein localization [31]. Furthermore, 3’UTRs also undergo splicing, which may affect transcript localization and the availability for translation [30]. Apart from that, specific pre-existing genetic abnormalities may predispose susceptible individuals to develop AML. A study by Brandsma et al. (2021) showed that differing mutational signatures could impact patient survival outcomes in pediatric AML [32]. Specifically, in the non-coding regions, single nucleotide polymorphisms (SNPs) are commonly observed, where 90% of single nucleotide polymorphisms (SNPs) are found in non-coding regions [33,34]. SNPs affecting the UTRs can potentially influence gene expression and cause cancer [35].

Numerous studies have indicated that 3’UTR shortening [29], lengthening [36] and/or mutations [37,38,39] play essential roles in solid cancer etiology and treatment; however, much remains unknown about their potential roles in AML. In this review, we will discuss the effects of dysregulation of circRNAs in AML recently and also highlight the roles that variations in 3’UTRs play in leukemogenesis.

## 2. CircRNAs

CircRNAs are covalently closed structures that are generally more stable than linear RNAs as they are resistant to exonuclease activity [40]. Most circRNAs are formed from the back-splicing of exons of pre-mRNAs [41]; some circRNAs also contain introns and/or intergenic sequences [42]. CircRNAs localize to the cytoplasm, nucleus or both and can also be transported by extracellular vesicles [43]. The expression levels of circRNAs are altered in many solid tumor types and hematopoietic malignancies [44]. Several earlier articles [27,42,45,46] have reviewed the known functions of circRNAs in AML. However, there have been a large number of research articles published since. Here, we will describe the latest findings on the biological mechanisms and functional consequences of circRNA dysregulation in AML.

### 2.1. Cellular Localization and Mechanisms of Action of CircRNAs

#### 2.1.1. Transcriptional/Epigenetic Regulators in the Nucleus

Some circRNAs, such as circATAD1 and circPLXNB2, are predominantly found in the nucleus [47,48] (Table 1). CircATAD1 is a nucleus-specific circRNA that is highly expressed in the bone marrow of AML patients. It is postulated to act as an upstream regulator of miR-34b by increasing methylation of the miR-34b [47]. Another predominantly nuclear circRNA, circPLXNB2, was shown to upregulate mRNA and the protein levels of PLXNB2 upon its overexpression, perhaps via increasing the transcription of its parental gene, PLXNB2 [48]. 

#### 2.1.2. Sponge for miRNAs and Binding to mRNAs in the Cytoplasm

Many cytoplasmic circRNAs act as miRNA sponges [49,50,51,52,53,54,55,56,57,58,59,60,61,62,63] (Table 1). CircRNA:miRNA interactions can be predicted using public databases such as CircNet [68], Circular RNA Interactome [69], CircBank [70] and starBase [71]. After bioinformatic predictions, validation methods such as RNA pulldowns [57] and luciferase assays in combination with mutational analysis [62] have led to the identification of several circRNAs, e.g., circ_0009910, circRNF220 and circ_0004136 as miRNA sponges.

Circ_0009910 is an exonic circRNA derived from exons 2 and 3 of the Mitofusin-2 (MFN2) gene, which is upregulated in AML patients [72]. It acts as a sponge of miR-5195-3p which subsequently affects the expression of its downstream target, GRB10 [60]. Circ_0009910 knockdown also resulted in lower B-cell lymphoma 2 (BCL-2) and higher Bcl-2-associated X protein (BAX) levels [60]. Other studies showed that circ_0009910 functions as a sponge for miR-20a-5p [72] and miR-491-5p [56] in AML. Circ_0009910 downregulates miR-491-5p and modulates the expression of B4GALT5, activating the PI3K/AKT signaling [56]. Additionally, all three studies indicated that circ_0009910 was associated with cell proliferation and inhibited apoptosis through these respective miRNAs (Figure 1).

CircRNF220, also known as circ_0012152, is highly expressed in the different subtypes (M1 to M5) of pediatric AML and it comprises exon 2 of the RNF220 gene [57]. This circRNA acts as a sponge for three different miRNAs: (1) miR-30a, which regulates MYSM1 and IER2 [57], (2) miR-330-5p, which modulates SOX4 expression [49] and (3) miR-625-5p, which controls SOX12 in AML cells [59]. All these miRNA:circRNF220 interactions were shown to inhibit apoptosis (Figure 1).

In addition, circ_0004136 RNA was shown to function as a sponge for miR-142 [73] and miR-570-3p [61] to enhance cell proliferation and tumor progression in pediatric AML (Figure 1).

Aside from indirect regulation of mRNAs via miRNA sponging, circRNAs can also bind directly to mRNAs. For example, hsa-circ_0003420 negatively regulates the mRNA of insulin-like growth factor 2 mRNA-binding protein (IGF2BP1) by targeting its 3’UTR (Table 1) to induce cell apoptosis [64].

#### 2.1.3. Protein Regulator in the Cytoplasm

Most circRNA expression levels correlate with the levels of their linear parental genes in both AML patients and healthy controls [74]. However, the expression pattern of circSPI1 was inverse to its parental gene SPI1. It was reported that circSPI1 was upregulated in AML patients, but SPI1 is a known tumor suppressor with decreased expression in AML [62]. Although circSPI1 had no impact on the mRNA expression of SPI1, it affected the protein level of SPI1 by interacting with the translation initiation factor eIF4AIII [62], suggesting that circSPI1 may directly regulate SPI1 protein expression.

#### 2.1.4. Extracellular Transport via Exosomes

CircRNAs are not only present in cells but can also be transported out of the cells via extracellular vesicles, such as exosomes [75]. Exosomes contribute to cellular communication during tumorigenic processes [76,77]. Therefore, exosome-derived circRNAs may represent potential cancer diagnostic markers and therapeutic targets [78]. Although exosomal circRNAs have been studied extensively in multiple cancers [79], little is known about their role in AML [46]. In 2021, two circRNAs, circ_0009910 [60] and circ_0004136 [61], were observed to be shuttled by exosomes. Circ_0009910 was upregulated in AML bone marrow and in 110 nm exosomes from both HL-60 and MOLM-13 cells [60]. Another circ_0004136 was detected in exosomes isolated from AML serum and HL-60 and K-562 cell lines [61]. Moreover, this mainly cytoplasmic circRNA was shown to act as a sponge for the tumor-suppressive miR-570-3p [80] and modulate the expression of the oncogenic driver TSPAN3 [81]. Indeed, these findings not only highlight the untapped potential of circRNAs as novel biomarkers but reveal the functional relevance of these exported circRNAs.

### 2.2. Biological Roles of CircRNAs

#### 2.2.1. Regulation of Cell Cycle Progression, Proliferation, Apoptosis and Metabolism 

Hematopoietic precursor cells (HPCs) in AML could proceed through limitless numbers of cell cycles and be resistant to cell death due to a multistep transformation process. Cell cycle control contributes to increased proliferation in AML and favors genomic instability. Additionally, hyperproliferation bears an enhanced risk of genetic damage and induces leukemogenesis [82]. Apoptosis, which is a programmed cell death mechanism, is critical for the development and homeostasis of the hematopoietic system. AML is known to evade apoptosis to favor cell survival and growth [83]. Metabolism is a series of dynamic processes that allow energy production according to cellular demands. Therefore, it is dependent on the cell’s stage of proliferation, differentiation and quiescence [84]. The rapidly proliferating AML cells require additional sources of energy for growth and survival [85]. The majority of the circRNAs studied to date have been shown to control cell cycle progression, proliferation, apoptosis and metabolism (Table 1 and Figure 1), such as circRNF220, which is highly expressed in pediatric AML patients [57] and in AML patient serum [49]. CircRNF220 knockdown in THP-1 and HL-60 cell lines also suppressed glucose consumption and lactate production, suggesting that it regulates glycolytic metabolism to facilitate AML progression (Figure 1) [49,86,87]. Similarly, the circ-SFMBT2/miR-582-3p/ZBTB20 axis was proposed as a regulator to induce cell proliferation, inhibit apoptosis and modulate glycolytic function in AML [54]. A recent study has discovered SIRT5 as a critical factor in multiple metabolic pathways, including in glycolysis capacity, OXPHOS, mitochondrial superoxide and glutamine metabolism which are required for leukemic cell survival [88]. 

Another study described that the upregulation of circTASP1 in the peripheral blood of AML patients correlated with reduced apoptosis and increased cell proliferation. This leukemogenic mechanism occurs through modulation of the miR-515-5p (a suppressor of breast and lung cancers)/HMGA2 (an oncogenic member of the high mobility group superfamily) axis [65].

#### 2.2.2. Regulation of Cell Phenotype or Differentiation

In a normal hematopoietic system, differentiation is tightly linked to cell proliferation [89]. Strong proliferation signals from cytokines lead to a G1 phase of cell cycle shortening and differentiation suppression [90]. Blocking the differentiation of immature blast cells to mature myeloid cells is a major driver of AML [2]. This differentiation block is thought to be driven by chromosomal translocations and mutations that affect transcription factors controlling differentiation [2]. Recently, studies have demonstrated that circRNAs could potentially modulate blood cell differentiation. For instance, hsa-circ_0003420 is lowly expressed in non-M3 AML stem cells. Overexpression of hsa-circ_0003420 in KG-1a cells causes leukemic stem cell (LSC) death and inhibits mRNA expression of the ABCB1 transporter, CD34 and MMRN1, which are the three key indicators of the LSC phenotype. Consistent results were also observed in flow cytometry (FC), where CD34+ and CD38- proportions were reduced by hsa-circ_0003420 overexpression [64]. Another circRNA, circRNF220, is upregulated in pediatric AML patients and enhances cell proliferation, the production of immature cells and downregulates granulocytic differentiation (CD11) and monocytic cell differentiation (CD14) [57]. Besides that, circSPI1 is also associated with cell differentiation. Silencing circSPI1 significantly induced granulocytic and monocytic differentiation in THP-1 and NB4 cells [62] (Table 1 and Figure 1).

#### 2.2.3. Regulation of Migration and Invasion

Circ_0004277 is downregulated in the bone marrow of AML patients, KG1a, THP-1, K-562 and U937 cells. It was shown to inhibit proliferation, migration and invasion in KG-1a and U937 cell lines via the miR-134-5p/single-stranded DNA binding protein 2 (SSBP2) axis [51]. On the other hand, circRNF220 promotes AML cells invasion by targeting miR-330-5p to increase the expression of SOX4 [49]; hsa_circ_0003602, miR-502/5p and IGF1R might form a regulatory axis to affect cell migration and invasion [52]. Circ-SFMBT2, circNPM1, circ_0004136 and circPLXNB2 were also shown to be associated with migration and invasion in AML cells, and their mechanisms are described in Section 2.2.1 (circ-SFMBT2), Section 2.3.2 (circNPM1) and Section 2.1.4 (circ_0004136). The mechanism is still unknown in cirPLXNB2 to induce migration and invasion in AML (Table 1 and Figure 1) [48,54,55,61]. 

### 2.3. Clinical Relevance of CircRNAs

#### 2.3.1. Prognostic, Diagnostic, Relapse and Subtype Markers

Many circRNAs have been explored as potential prognostic, diagnostic and relapse markers in AML (Figure 2). For example, has_circ_0079480 was shown to be upregulated in the serum of AML patients compared to that of healthy controls by qRT-PCR. Among different AML subtypes, has_circ_0079480 was highly expressed in the M4/M5 type. The upregulation of this circRNA is correlated with a worse overall survival rate (OS) and relapse-free survival rate (RFS) of AML patients. Furthermore, has_circ_0079480 expression levels were decreased after treatment and in patients that achieved complete remission. Based on these results, has_circ_0079480 may have a prognostic utility in AML [66].

Likewise, circPLXNB2 could also be a prognostic predictor for AML as it is highly expressed in AML patients. The OS was higher in patients with lower circPLXNB2 expression compared to patients with higher circPLXNB2 expression [48]. Another circRNA, circ-ANAPC7, appeared to be upregulated in AML through microarray analysis and qRT-PCR [91]. A follow-up study showed that circ-ANAPC7 expression was higher in the AML group compared to iron deficiency anemia (IDA), which is the control group, and CR samples. Circ-ANAPC7 may thus serve as a potential diagnosis and relapse biomarker in AML [67]. Similarly, circRNF220 is overexpressed in pediatric patients and decreased dramatically in CR patients, who showed no difference in circRNF220 expression compared to the control group. In an additional experiment involving 87 primary AML patients, circRNF220 expression was approximately 2.5-fold higher in relapse samples (n = 11) compared to the CR patients (n = 76). The expression of circRNF220 was only highly expressed in AML but not in acute lymphoblastic leukemia (ALL) or other hematological diseases [57]. Taken together, circRNAs are potential biomarkers for monitoring the disease status and therapeutic response.

#### 2.3.2. Drug Resistance

As mentioned earlier, the CR rate of the first-line treatment for older AML patients is 40–60%. However, nearly 60% of elderly patients with recurrence fail to respond to chemotherapy. Recent studies have suggested that treatment failure was due to drug resistance [3]. Gene mutations, the aberrant expression of miRNAs, upregulated PI3K/AKT and autophagy signaling pathways and drug resistance-related enzymes have all been implicated in patient relapse and drug resistance [3]. An FLT3 inhibitor treatment study demonstrated that FLT3 (26%), epigenetic modifiers (16%), RAS/MAPK pathway genes (13%), WT1 (7%) and TP53 (7%) are emerging mutations identified in relapsed AML patients [8]. In addition to gene mutations, current studies have also linked circRNAs to drug resistance. For example, circPAN3 was upregulated in doxorubicin (ADM)-resistant THP1 cells. The circPAN3-miR-153-5p/miR-183-5p-XIAP axis and the circPAN3-mediated autophagy regulation via the AMPK/mTOR pathway are potentially important mediators of drug resistance in AML patients [92,93]. Similarly, circNPM1 expression was upregulated in AML patients and contributed to Adriamycin (ADM) chemoresistance by targeting miR-345-5p and regulating FZD5 expression. These circRNAs may represent therapeutic vulnerabilities in drug-refractory AML [55] (Figure 2).

## 3. 3’ Untranslated Regions (3’UTRs) of mRNAs

Recently, studies have shown that circRNAs can bind to 3’UTRs and directly regulate tumorigenicity. For example, circZNF609 controls the stability and translational activity of CKAP5 by binding to its 3’UTR. Consequently, the upregulation of CKAP5 leads to increased cell proliferation in tumorigenesis [94]. Specifically in AML, has-circ_0003420 targets the 3’UTR of IGF2BP1 as described in Section 2.1.2 [64]. Interestingly, studies have also shown that circRNAs can sequester miRNAs from other target transcripts, thus acting as competing endogenous RNAs (ceRNAs) in AML [95]. Considering that the majority of known miRNA target sites are located in 3’UTRs, it is necessary to comprehensively map the landscape of miRNA sites in 3’UTRs to fully understand the extent of circRNA ceRNA function. Additionally, 3’UTRs contain many other regulatory elements that control cognate gene expression. Although they may not be studied as extensively as the protein-coding regions of mRNAs, recent studies have begun to shed light on how 3’UTR heterogeneity and functions may contribute to leukemogenesis. 

### 3.1. APA within 3’UTRs

Current studies have begun to reveal the previously uncharacterized roles of the non-coding regions of protein-coding RNAs, or UTRs, in AML. The shortening of 3’UTR by APA has been shown to contribute to tumorigenesis [96,97,98,99] and leukemia development (Figure 3). In recent studies, CPSF1 and FIP1L1, components of the cleavage and polyadenylation factors complex, have been demonstrated to regulate the 3’UTR length of fusion transcripts. In AML, aberrant chromosomal translocations can generate gene fusions, producing oncofusion proteins which underpin tumorigenesis. One of the most widely detected oncofusion genes, AML1-ETO (AE), is widely known to be responsible for myeloid leukemogenesis [100,101,102]. Shima et al. (2019) showed that the AE fusion gene consists of four canonical PASs and favors a shorter length in both t(8;21) AML primary patients and cell lines [103]. To investigate the key regulator of the observed 3’UTR changes, they knocked down all APA machinery members and found that CPSF1 promotes AE 3’UTR shortening to increase AE stability and confer growth advantage to AML cells.

Likewise, FIP1L1 regulates APA favoring shorter 3’UTR lengths and reduces cell differentiation by attenuating the mechanistic target of rapamycin complex 1 (mTORC1) signaling and modulating MYC protein levels [104]. Indeed, it is essential to investigate how APA could play a pivotal role in leukemogenesis and explore potential targeted therapeutic approaches.

### 3.2. 3’UTR Splicing

To date, most cancer research studies have focused on alternative splicing (AS) in protein-coding regions. Although 3’UTR splicing events are annotated in databases, little is known about their potential functions in physiological and pathophysiological conditions. Recently, a pan-cancer study revealed that 3’UTR splicing is widespread, significantly upregulated in multiple cancer types including AML [30] and associated with poor patient outcomes. The top candidate, 3’UTR splicing of CTNNB1, was over-spliced in approximately 40% of solid tumor samples and AML. The CTNNB1 3’UTR splice variant was shown to promote cell proliferation and migration in hepatocellular carcinoma (HCC) and colon adenocarcinoma (COAD). Mechanistically, 3’UTR splicing increases the cytoplasmic localization of the transcripts and this differential localization of CTNNB1 3’UTR variants could be a key factor modulating CTNNB1 protein expression. Although 3’UTR splicing of certain genes such as CTNNB1 is conserved between AML and solid cancers, it is intriguing to note that the majority of 3’UTR splicing events are not conserved between blood and solid tumors. Further studies will be crucial to shed light on the functional implications of 3’UTR splicing in AML and evaluate potential diagnostic, prognostic and therapeutic applications.

### 3.3. SNPs within 3’UTRs

SNPs are generally found in non-coding regions and can result in heterogeneous gene expression and the subsequent alterations of transcript and/or protein functions [105]. Several studies have shown that specific SNPs are associated with favorable outcomes for AML patients [106,107]. On the contrary, there are also reports of SNPs giving rise to aberrant gene expression which leads to adverse outcomes and worse survival in AML patients. For example, a polymorphic deletion of a ‘T’ nucleotide in the 3’UTR of NPM1 was reported in 60-70% of AML patients [108,109]. This T deletion in the 3’UTR facilitates the illegitimate binding of miR-337-5p, which is highly expressed in different AML subtypes and reduces the expression of NPM1 (Figure 3) [110].

Another study identified an SNP homozygous AA genotype located within the 3’UTR of ABCB1, which increases its product, P-glycoprotein (p-gp), expression compared to GA and GG genotypes in AML patients (Figure 3) [111]. Luciferase reporter assays showed that the A allele in the 3’UTR of ABCB1 RNA was associated with higher expression of p-gp expression. High levels of p-gp expression have been widely reported to be associated with clinical therapy failure or drug resistance [112,113]. In summary, polymorphisms in 3’UTRs could regulate the expression of cognate gene expression and play important roles in AML pathogenesis.

## 4. Conclusions

In closing, we have summarized the recent advancements of circRNAs in clinical usage and the roles of circRNAs and 3’UTRs in AML pathogenesis. Among these, circRNAs are the most well-studied and usually use the loss of function (LoF) method to study their cellular and biological roles in AML. LoF studies generally utilize an RNA interference (RNAi)-based method with short hairpin RNAs (shRNAs) or small interfering RNAs (siRNAs) to target the back-splice junction [114]. Since circRNAs have the same sequence as their parental genes, the partial complementarity of an RNAi would bind to the linear mRNA. In recent years, Guarnerio et al. (2019) used LNA GapmeRs to minimize the off-targeting to linear cognate RNAs by transfection [115] and Li et al. (2021) reported that CRISPR-Cas13 with the lentivirus system achieves robust circRNA-specific knockdown without disturbing cognate mRNAs [116]. These two methods have not been utilized in AML and they might be useful in LoF studies of circRNA in the future.

Regardless of the limitation in LoF studies of circRNAs, many physiological roles of circRNAs have been uncovered. CircRNAs have well-documented roles as miRNA sponges and regulate cell cycle progression, proliferation and apoptosis in AML (Table 1 and Figure 1). However, the roles of circRNAs are not restricted to the aforementioned mechanism. Research in solid cancers has demonstrated that circRNAs have diverse cellular roles in regulating transcription and splicing in the nucleus, the translational regulation in the cytoplasm through the interaction with mRNA/protein and the downstream control of protein expression. Therefore, it would be critical for future studies to holistically examine all the interacting partners, e.g., mRNA and RBPs, with their associated circRNAs and regulatory networks in both the nucleus and cytoplasm [117]. These cellular functions of circRNAs can be investigated in the context of AML in the future and utilized clinically.

Relapse occurs in approximately 40–50% of patients after receiving standard treatment [118]. Various alternative novel therapies have been introduced to these relapse patients such as a BET inhibitor (BETi) that targets the AHR signaling [119] and Ficlatuzumab [120], a first-in-class anti-HGF antibody that inhibits the MET signaling [121]. Currently, there are several studies that suggest circRNAs as potential non-invasive biomarkers for diagnosis, prognosis, relapse and subtype classification. However, limited research has explored the potential utility of circRNA as RNA therapeutics or novel treatment options in AML. Further works will likely reveal their prospect as novel RNA-based therapeutics, thus providing more treatment alternatives for relapse patients.

Similarly, although there is increasing evidence for the dysregulation of 3’UTRs in AML, the underlying molecular mechanisms and their functional implications remain poorly understood. The heterogeneity of 3’UTRs may have a profound impact on the landscapes of bound miRNAs and RBPs, which in turn will affect the regulatory RNA networks encompassing circRNAs and other RNA species. Future studies in these areas may pave the way to unlock their potential for novel theragnostic applications.

## Figures and Tables

**Figure 1 ijms-24-03215-f001:**
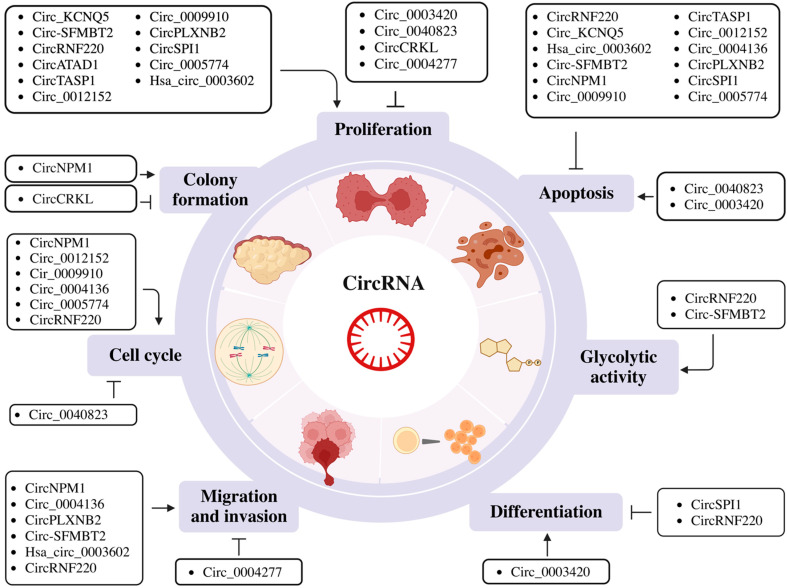
Schematic representation of the biological role of circular RNAs (circRNAs) in proliferation, apoptosis, glycolytic activity, differentiation, migration and invasion, cell cycle regulation and colony formation. Figures were created with BioRender.com.

**Figure 2 ijms-24-03215-f002:**
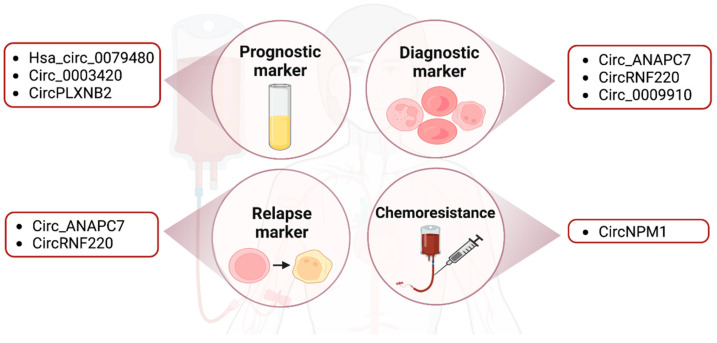
The clinical relevance of circRNAs in AML. Figures were created with BioRender.com.

**Figure 3 ijms-24-03215-f003:**
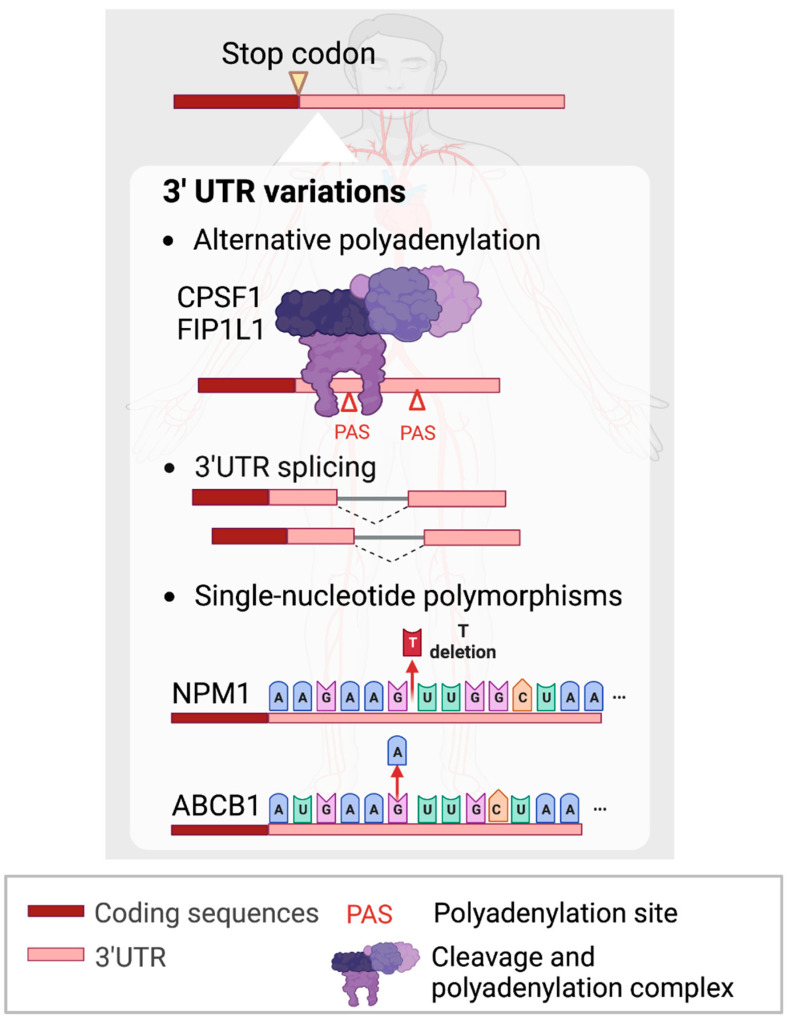
Non-coding 3’UTR variations contributing to leukemogenesis. Figures were created with BioRender.com.

**Table 1 ijms-24-03215-t001:** Circular RNAs in acute myeloid leukemia (AML).

Non-Coding RNA	Host Gene	Localization	Sample and Cell Lines/Expression Pattern	Target miRNA/mRNA or Protein	Function	Ref.
CircATAD1	ATAD1	Nucleus	Up in BM	miR-34b	Induces proliferation and miR-34b methylation.	[47]
CircPLXNB2	PLXNB2	Nucleus	Up in AML patients, OCI-AML3 (Not in HL-60, THP1, MV4-11)		Prognostic marker. Induces cell proliferation and migration. Inhibits apoptosis.	[48]
CircRNF220	Ring Finger Protein 220	Cytoplasm	Up in serum, THP-1 MV4-11, KG-1, HL-60 and AML2	miR-330-5p/SOX4	Induces cell progression, invasion and glycolytic metabolism. Inhibits apoptosis.	[49]
Circ_KCNQ5	KCNQ5	Cytoplasm	Up in BM from childhood patients, Kasumi-1 and MV4-11	miR-622/RAB10	Induces proliferation and inhibits cell apoptosis.	[50]
Circ_0004277		Cytoplasm	Down in BM, THP-1, K-562, U937 and KG-1a	miR-134-5p/SSBP2	Inhibits cell viability, migration and invasion.	[51]
Hsa_circ_0003602			Up in BM, HL60, K-562 and THP-1	miR-502-5p/IGF1R	Induces cell proliferation, migration and invasion. Inhibits apoptosis.	[52]
Circ_0040823			Down in peripheral blood, HL-60, THP-1, U937 and KG-1	miR-516b/PTEN	Inhibits proliferation. Induces apoptosis and cell cycle arrest.	[53]
Circ-SFMBT2	SFMBT2	Cytoplasm	Up in serum, U937, NB4, K-562 and HL-60	miR-582-3p/ZBTB20	Induces proliferation, migration, invasion and glycolysis. Inhibits apoptosis.	[54]
CircNPM1	NPM1	Cytoplasm	Up in serum, THP-1 and HL-60	miR-345-5p/FZD5	Induces colony formation, migration, invasion and Adriamycin resistance. Inhibits apoptosis and cell cycle arrest.	[55]
Circ_0009910	MFN2		Up in BM, HL60 and MOLM-13	miR-491-5P/B4GALT5 and PI3K/AKT signaling pathway	Induces proliferation, sphere formation and autophagy. Inhibits apoptosis.	[56]
CircRNF220	RNF220	Cytoplasm	Up in BM of pediatric patients and primary AML cells	miR-30a	Diagnosis, prognosis and relapse markers. Induces proliferation. Inhibits apoptosis and regulates cell differentiation.	[57]
CircCRKL	CRKL	Cytoplasm	Down in BM, KG-1a, KG-1, Kasumi-1, THP1, MOLM-13	miR-196a-5p and miR196b-5p/p27	Inhibits proliferation and colony-forming ability.	[58]
Circ_0012152	RNF220		Up in BM, U937 and HL60 cells	miR-625-5p/SOX12	Induces proliferation. Inhibits apoptosis and regulates cell cycle.	[59]
Cir_0009910		Cytoplasm and shuttled by exosome	Up in BM, HL-60 and MOLM-13	miR-5195-3p/GRB10	Potential diagnostic marker for myeloid leukemia. Induces proliferation. Inhibits apoptosis and regulates cell cycle.	[60]
Circ_0004136	KCNQ5	Cytoplasm and shuttled by exosome	Up in serum of pediatric AML patients, HL60 and K-562	miR-570-3p/TSPAN3	Induces cell viability, migration and invasion. Inhibits apoptosis and regulates cell cycle.	[61]
CircSPI1	SPI1	Cytoplasm	Up in AML patients, THP1 and NB4	miR-1307-3p, miR-382-5P, miR-767-5p	Induces proliferation. Inhibits myeloid differentiation and apoptosis.	[62]
Circ_0005774			Up in blood of pediatric AML, HL-60 and NB4	miR-192-5p/ULK1	Induces proliferation. Inhibits apoptosis and regulates cell cycle.	[63]
Circ_0003420 (Hsa_circ_0003420)			Down in non-m3 AML stem (BM) cells	3’UTR of IGF2BP1	Prognostic marker. Inhibits leukemogenesis. Induces apoptosis.	[64]
CircTASP1 (Hsa_circ_0007340)	TASP1	Cytoplasm	Up in peripheral blood, HL60, KG-1, U937 and THP-1	miR-515-5p/HMGA2	Induces proliferation. Inhibits apoptosis.	[65]
Hsa_circ_0079480			Up in serum and most profound in M4/M5 type		Prognostic marker.	[66]
Circ-ANAPC7	ANAPC7		Up in BM		Diagnosed and relapsed AML.	[67]

Abbreviations: Circ, circular RNAs; BM, bone marrow; up, upregulated in AML patients; down, downregulated in AML patients. AML cell lines: AML2, HL-60, KG-1, KG-1a, MOLM-13, MV4-11, NB4, OCI-AML3, THP-1 and U937; CML cell line: K-562.

## Data Availability

Not applicable.

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
