# Peer review of "Circular RNAs and Untranslated Regions in Acute Myeloid Leukemia"

_ijms, 2023, doi:10.3390/ijms24043215_

Round 1
Reviewer 1 Report
The review intitituled Non-Coding RNAs And Untranslated Regions in Acute Myeloid Leukemia, that the authors in the abstract and the intoduction part discuss the effects of dysregulation of different ncRNA species in AML and also highlight the roles that variations in 3’UTRs play in leukemogenesis.
However, they devoted six pages for circular RNAs and less than two pages for untranslated regions (3'UTRs) of mRNAs and half a page for miRNAs and LncRNAs.
A very poor review of the subject and the title does not reflect the content of the article The organization needs to be strongly reworked
The discussion part is very poor in content
For Introduction part, some contents needs to be reworked for example :
Acute myeloid leukemia (AML) is characterized by clonal expansion of immature blast cells (>20%) in the bone marrow and peripheral blood [1],
The new caracterization of AML is not just up to 20%.
https://aml-hub.com/medical-information/2022-eln-recommendations-for-the-diagnosis-of-aml-in-adults
While the blast threshold remains at 20% for the majority of AML subtypes, the recurrent genetic lesions, are now considered to indicate AML when there are ≥10% bone marrow (BM) or blood blasts.
Line 29 Most AML patients achieve complete remission (CR)
To add percent and to change CR on first complete remission
The CR rate of first-line treatment is 60%-80% in younger patients (<65 years old) and 31 40%-60% in patients over 65 years old [4-6].
Need to be clarified in totality just 30% achieve CR
It is important to discuss the role of Piwi-interacting RNA and the other ncRNA
It is important to note the other mechanisms described in the literature by which miRNAs regulate gene expression
I suggest that the article be reworked focusing on circular RNAs and untranslated regions (3'UTRs) of mRNAs and the relationship between the two in the pathogenesis of LAM
Reviewer 2 Report
In this review, the authors summarized recent advances about the role non-coding RNA and untranslated region in pathogenesis diagnosis and therapy of AML. The manuscript covers a wide spectrum of the regulation mechanism and biological function for non-coding RNA and untranslated region. The review is comprehensive, while some important issues need to be addressed to improve the manuscript.
1. The title of this manuscript is “Non-Coding RNAs And Untranslated Regions in Acute Myeloid Leukemia”. Non-coding RNA contains lncRNA, miRNA, circRNA, etc. However, most of the manuscript discussed about circRNA, and only two paragraphs described antisense-RNA and enhancer RNA. Thus, it is better to delete these two paragraphs and change “Non-Coding RNAs” in the title into “CircRNA”.
2. In section 2.2.1, how circ-SFMBT2 regulated cell progress is not well presented.
3. Section 2.2.3 should be extended to present the mechanism of how these circRNA regulating cell migration and invasion rather than simply listing.
4. Some references are too old. References published within the last three years are encouraged to be updated in the revised version. Here I provide some excellent references for the authors: 1) Monocytic differentiation and AHR signaling as Primary Nodes of BET Inhibitor Response in Acute Myeloid Leukemia [PMID: 34568834]; 2) Mutation signatures of pediatric acute myeloid leukemia and normal blood progenitors associated with differential patient outcomes [PMID: 34642666]; 3) Patterns of Resistance Differ in Patients with Acute Myeloid Leukemia Treated with Type I versus Type II FLT3 inhibitors [PMID: 33681815]; 4) SIRT5 Is a Druggable Metabolic Vulnerability in Acute Myeloid Leukemia [PMID: 34027418]; 5. Inhibition of MET Signaling with Ficlatuzumab in Combination with Chemotherapy in Refractory AML: Clinical Outcomes and High-Dimensional Analysis [PMID: 34514432].
5. The font in tables and figure legends is not consistent.
Round 2
Reviewer 2 Report
The Authors have addressed all of my concerns.